# Investigation of Therapeutic Efficacy of Intravesical Tigecycline Administration in Rats with Cystitis Induced by Extensively Drug-Resistant (XDR), Tigecycline-Sensitive *Acinetobacter baumannii* Strain

**DOI:** 10.3390/antibiotics14060611

**Published:** 2025-06-16

**Authors:** Cihan Yüksel, Işıl D. Alıravcı, Murat Koşan, Sinem Esen, Sevinç Yenice Aktaş, Neslihan Kaya Terzi, Ahmet Ali Berber, Sevil Alkan, Selçuk Kaya

**Affiliations:** 1Department of Infectious Diseases and Clinical Microbiology, Faculty of Medicine, Çanakkale Onsekiz Mart University, 17000 Çanakkale, Turkiye; cihan.yuksel3@saglik.gov.tr (C.Y.); sevil.alkan@comu.edu.tr (S.A.); selcuk.kaya@comu.edu.tr (S.K.); 2Department of Urology, Faculty of Medicine, Çanakkale Onsekiz Mart University, 17000 Çanakkale, Türkiye; mkosan@comu.edu.tr; 3Department of Medical Microbiology, Faculty of Medicine, Çanakkale Onsekiz Mart University, 17000 Çanakkale, Türkiye; sinem.esen1@saglik.gov.tr (S.E.); sevinc.yeniceaktas@comu.edu.tr (S.Y.A.); 4Department of Medical Pathology, Faculty of Medicine, Çanakkale Onsekiz Mart University, 17000 Çanakkale, Türkiye; neslihan.kayaterzi@comu.edu.tr; 5Vocational School of Health Services, Çanakkale Onsekiz Mart University, 17000 Çanakkale, Türkiye; aberber@comu.edu.tr

**Keywords:** XDR, *Acinetobacter baumannii*, tigecycline, intravesical therapy, genotoxicity, rat model

## Abstract

Background: This study aimed to evaluate the therapeutic efficacy of intravesical tigecycline administration in a rat model of cystitis induced by a tigecycline-sensitive, extensively drug-resistant (XDR) *Acinetobacter baumannii* strain. Methods: Thirty-six female Wistar albino rats were inoculated intravesically with XDR *A. baumannii* to induce cystitis. Twenty-four rats that developed infection were divided into four groups: untreated control, saline irrigation, low-dose tigecycline (6.25 mg/kg), and high-dose tigecycline (25 mg/kg). Microbiological clearance was assessed via urine cultures on days 3 and 5. Bladder tissues were analyzed histopathologically and for genotoxicity using the Comet assay. Results: On day 5, microbiological clearance was significantly higher in tigecycline-treated groups compared to controls (*p* = 0.028). Histopathology revealed significantly more inflammation in the high-dose tigecycline group (*p* = 0.029). Genotoxicity was observed in both tigecycline groups, independent of dose (*p* < 0.05). Conclusions: Intravesical tigecycline demonstrated microbiological efficacy against XDR *A. baumannii*-induced cystitis. However, its inflammatory and genotoxic potential necessitates further preclinical evaluation.

## 1. Introduction

Cystitis caused by extensively drug-resistant (XDR) *Acinetobacter baumannii* has emerged as a significant therapeutic challenge, particularly among patients with neurogenic bladder and those requiring frequent urinary catheterization or clean intermittent catheterization (CIC). In such populations, the structural and functional abnormalities of the lower urinary tract or neurological conditions such as spinal cord injury (SCI), multiple sclerosis, cerebrovascular events, and Parkinson’s disease predispose to recurrent urinary tract infections (UTIs), often caused by multidrug-resistant organisms for which systemic antibiotic therapy may be ineffective or poorly tolerated [1,2].

Intravesical antibiotic administration is increasingly employed as a localized treatment strategy for UTIs. This approach intends to circumvent the gastrointestinal tract and reduce systemic adverse effects, including gastrointestinal disturbances and antibiotic-associated colitis. Intravesical therapy is generally delivered via transurethral catheterization, with the antibiotic retained in the bladder for a defined period to maximize local antimicrobial exposure. However, there are ongoing concerns regarding increased systemic absorption in the setting of mucosal inflammation, vesicoureteral reflux, or in immunocompromised patients such as kidney transplant recipients [3,4,5,6].

Several studies have explored strategies for preventing recurrent UTIs in patients with impaired bladder emptying, including intravesical antibiotic therapy. Previous studies have already tried different antibiotic options generally focused on the use of gentamicin [4,5,7,8,9,10,11,12,13]. Among available agents, gentamicin is the most extensively studied and has demonstrated both prophylactic and therapeutic efficacy against UTIs, particularly those caused by multidrug-resistant Gram-negative bacteria [5,13,14,15,16].

Treatment options for carbapenem-resistant *A. baumannii* (CRAB) are limited, and resistance to available antibiotics is increasing. For CRAB infections, combination regimens involving at least two agents are often necessary, which may contribute to renal and hepatic dysfunction in patients [17]. Tigecycline, a semi-synthetic tetracycline derivative and glycylcycline antibiotic, inhibits protein synthesis at the ribosomal level and demonstrates broad-spectrum antibacterial activity [18]. It remains a valuable agent against MDR and XDR A. baumannii isolates, with resistance rates lower than those reported for carbapenems. Despite its clinical relevance, tigecycline achieves minimal urinary concentrations following systemic administration, limiting its utility in treating UTIs through conventional routes and systemic intravenous administration is not recommended for UTIs [19,20,21].

Although tigecycline is widely used for the treatment of resistant bacterial infections, recent experimental studies have raised concerns about its potential genotoxic effects, especially when administered at high doses. Proposed mechanisms include oxidative stress, mitochondrial dysfunction, and inhibition of DNA repair. Previous research has demonstrated increased DNA strand breaks and oxidative markers following tigecycline exposure in both in vitro and in vivo models [22,23,24].

Surveillance studies in Turkey have reported carbapenem resistance rates as high as 91% among A. baumannii isolates, while tigecycline resistance remains comparatively lower, ranging from 11% to 16% [25,26]. These findings underscore the potential utility of alternative administration routes, such as intravesical instillation, for enhancing the therapeutic efficacy of tigecycline in urinary tract infections.

To our knowledge, no prior in vivo experimental model of intravesical administration of tigecycline exists in the literature. This study aimed to investigate the therapeutic potential of intravesical tigecycline in a rat model of cystitis induced by a tigecycline-susceptible XDR A. baumannii strain. We hypothesized that this localized approach could offer a viable alternative to systemic therapy by enhancing drug exposure at the site of infection while minimizing systemic toxicity, hospital stay, and healthcare-associated costs.

## 2. Results

The results of our study are shown below under three subheadings.

### 2.1. Microbiological Findings

Urine culture results were evaluated on the 3rd day of treatment (7th day after inoculation) and the 5th day of treatment (9th day after inoculation) for each experimental group. The study groups included Group 1 (untreated control group), Group 2 (control group treated with saline solution), Group 3 (low-dose tigecycline treatment group), and Group 4 (high-dose tigecycline treatment group). The culture negativity rates were compared between the groups. On the 5th day of treatment, both the low-dose and high-dose tigecycline treatment groups demonstrated significantly higher microbiological clearance rates in urine compared to the control groups. High-dose tigecycline, in particular, achieved the highest culture negativity rate (83.3%). Thus, the treatment groups achieved statistically significant microbiological clearance in the urine compared to the control groups. While no statistically significant difference was observed on the 3rd day of treatment (*p* = 0.082), a statistically significant difference was found on the 5th day of treatment (*p* = 0.028) (Table 1).

Untreated Control Group (Group 1): All six rats (100%) showed bacterial growth in their urine cultures, and no rats were culture-negative on the third and fifth days.

Saline Solution Group (Group 2): While two rats (33.3%) showed no bacterial growth, four rats (66.7%) were urine culture-positive both on the third and fifth days.

Low-Dose Tigecycline Group (Group 3): While four rats (66.7%) had no bacterial growth, 2 rats (33.3%) were culture-positive both on the third and fifth days.

High-Dose Tigecycline Group (Group 4): While two rats (33.3%) were culture-positive on the third day, one rat (16.7%) showed bacterial growth on the fifth day.

### 2.2. Histopathological Findings

Bladder tissues obtained from rats were evaluated under light microscope for inflammation, fibrosis, vascular proliferation, edema, ulceration, erosion, and epithelial regeneration. Table 2 shows histopathological findings by analyzing the percentage of among the groups. There was significant difference in terms of inflammation among groups according to the *p* value, which was found to be 0.028. There was no significant difference among groups in terms of fibrosis, vascular proliferation, edema, ulceration, erosion, and epithelial regeneration (Table 2, Figure 1).

### 2.3. Cytogenetic Findings

Bladder tissues taken from rats were subjected to the Comet test after appropriate preparations. Tail length, tail moment, and tail density parameters were analyzed for each sample. Statistically significant differences were found between the treatment groups and control groups. Table 3 shows the comparison results of the tail length, tail moment, and tail density parameters calculated as a result of the Comet test between the groups. Figure 2 demonstreates the movement of comet cell on electrophoresis.

## 3. Discussion

XDR *A. baumannii* is a common pathogen responsible for catheter-associated urinary tract infections (CAUTIs) in intensive care unit (ICU) patients. Its treatment typically requires high-dose combination systemic antibiotics, which pose risks of renal and liver impairment [27].

Tigecycline is one of the few available treatment options for XDR *A. baumannii* infections; however, it is not typically used for urinary tract infections (UTIs) due to its low urinary concentrations when administered systemically. The absence of intravesical pharmacokinetic data is a limitation of our study. This limitation prompted us to explore its potential efficacy when delivered locally via intravesical administration. In our study, intravesical administration of the drug directly to the infection site demonstrates that systemic limitations can be overcome. Similarly, Song et al. reported that the local efficacy of colistin administered into the bladder in an animal model was higher than systemic administration [28]. Although numerous studies in the literature have examined intravesical treatments for various antibiotics no data exist regarding tigecycline [8,9,10,29,30,31]. However, our dose selection was informed by tolerability and precedent in related infection models, and future studies assessing bladder tissue concentrations and systemic absorption of intravesically administered tigecycline are warranted.

Animal studies involving tigecycline have primarily focused on its use in infected wounds, osteomyelitis, pneumonia, sepsis, and peritoneal dialysis solutions [32,33,34,35]; however, its effect on the urinary system has not been adequately evaluated. In 1987, McGuire et al. were the first to investigate the use of intravesical gentamicin for the treatment of recurrent urinary tract infections (UTIs), achieving successful outcomes [36]. Following this, Defoor et al. studied intravesical gentamicin, Welk et al. examined fosfomycin, Wan et al. also explored gentamicin, and Assis et al. investigated colistin using animal models of intravesical therapy. All of these studies reported successful results with these local treatment approaches [5,7,14,37]. For instance, in Goessens’ study on the therapeutic efficacy of tigecycline in experimental pneumonia caused by *Klebsiella pneumoniae* strains producing extended-spectrum beta-lactamases (ESBLs), rats were treated with tigecycline at doses of 6.25, 12.5, or 25 mg/kg twice daily for 10 days. The study reported a survival rate of over 90% at the highest dose [32]. In our study, we employed tigecycline at doses of 6.25 mg/kg (low dose) and 25 mg/kg (high dose) to evaluate its therapeutic efficacy in the treatment of bladder infections regarding to Goessens’ study, representing a therapeutic and a high-exposure level. Although pharmacokinetic data specific to intravesical use are lacking, these doses were well tolerated in preliminary pilot trials, and the five-day duration was chosen to approximate a typical antimicrobial treatment course in humans.

In this study, cystitis did not develop in seven rats despite standardized inoculation procedures, and five animals died during the experimental process. Possible explanations for the lack of cystitis include variability in bacterial adhesion and host immune response, or technical deviations during intravesical instillation. The deaths may have resulted from anesthesia-related complications, stress-related physiological responses, or underlying conditions not detected during randomization. While these exclusions may have affected group sizes, the statistical analyses retained sufficient power to detect significant differences, and the validity of the overall conclusions remains intact.

In this study, no statistically significant difference in microbiological clearance was observed between the treatment and control groups on day 3 of intravesical tigecycline administration (*p* = 0.082), whereas by day 5, this difference became statistically significant (*p* = 0.028). This finding suggests that longer treatment durations may be necessary for tigecycline to achieve effective eradication in localized bladder infections. In the literature, systemic tigecycline treatments are typically recommended for 7–14 days, and shorter durations have been associated with increased risk of recurrence [21]. However, data regarding the optimal duration of intravesical therapies are limited. Song et al. demonstrated that a 5-day intravesical colistin regimen achieved significant microbiological clearance in an experimental setting [28]. In this context, our findings indicate that 5 days may also represent a minimum effective duration for intravesical tigecycline therapy. Nonetheless, the presence of inflammation and potential genotoxicity underscores the need to carefully evaluate the safety of this duration.

In this study, pathological evaluation revealed significant inflammation in the bladder tissue, particularly in the high-dose intravesical tigecycline group (*p* = 0.029). This finding suggests a potential local irritant effect or dose-dependent cytotoxic response associated with tigecycline. Similar inflammatory effects of tigecycline on tissues have been reported in other experimental models. Zhang et al. demonstrated that high concentrations of tigecycline could induce cellular damage and inflammatory responses in human cell lines [38]. Furthermore, a study by Kaur et al. showed that high-dose local antibiotic administration can lead to edema, congestion, and inflammation in bladder epithelium in rats [39]. In light of these findings, our study suggests that high-dose intravesical tigecycline may provoke inflammatory changes in bladder tissue, highlighting the need for careful dose optimization. Future studies should explore lower-dose regimens to evaluate both therapeutic efficacy and tissue tolerability.

Another remarkable finding of this study was the increased inflammatory response observed in the high dose tigecycline group and the genotoxicity findings observed in both tigecycline groups regardless of dose. In the literature, there are many genotoxicity studies of antibiotics in use based on the measurement of DNA damage in cultured human lymphocytes [40,41,42,43,44]. There are studies in the literature showing that tetracycline has cytotoxic and genotoxic effects in human blood lymphocytes [41,45]. Previous studies have reported that tetracyclines induce oxidative stress and mitochondrial dysfunction, leading to accumulation of ROS and DNA strand breaks [23,46].

Importantly, the inclusion of non-infected groups receiving tigecycline enabled us to assess the drug’s direct genotoxic potential independent of infection-induced inflammation. The significant increase in DNA damage parameters in these groups supports the hypothesis that tigecycline, particularly at higher doses, may cause intrinsic genotoxic effects. These findings align with previous studies reporting DNA damage associated with tetracycline derivatives in non-infectious settings [22,23]. In a similar direction, Zhang et al. reported that high concentrations of tigecycline may be associated with DNA damage and apoptosis in some cell cultures [38]. Thus, infection-independent genotoxicity must be considered in the context of localized tigecycline applications, especially when used in immunocompromised or repeatedly exposed populations. Our study suggests the presence of direct cytotoxic effects due to local application of the drug. Since bladder tissues are analyzed immediately after the end of treatment, additional studies are needed to determine whether these cytotoxic and genotoxic effects persist in the long term.

## 4. Materials and Methods

Ethical approval for the study was obtained from the Çanakkale Onsekiz Mart University Animal Experiments Local Ethics Committee (decision number: 2024/01–04; date: 26 January 2024). All procedures complied with the “Regulation on the Welfare and Protection of Animals Used for Experimental and Other Scientific Purposes” (13 December 2011-28141) issued by the Ministry of Food, Agriculture, and Livestock. The study was conducted at Çanakkale Onsekiz Mart University Experimental Research and Application Center between August and September 2024.

### 4.1. Animals

An experimental model of infected rat bladders was created for the study. A total of 36 Wistar female rats, aged 10–12 weeks and weighing 250–300 g, were used. Prior to the study, all rats were confirmed to be healthy based on systemic (respiratory, cardiovascular) examinations. The rats were housed under standard conditions: 25 °C room temperature, 50–60% humidity, and a 12 h light–dark cycle. They were provided with city water and standard pellet feed ad libitum.

### 4.2. Bacterial Cystitis Model

The experimental study lasted nine days. On day 0, a clinical isolate of XDR tigecycline-sensitive *A. baumannii* (15 × 10^8^ CFU/mL, 5 McFarland) in 1 mL saline solution was inoculated into the bladders of the rats via transurethral catheterization and the cystitis model was established (Figure 3). *A. baumannii complex* strain was isolated from the blood culture of a 3-year-old male patient in the trauma emergency intensive care unit (clinical isolate from a tertiary hospital). To ensure sufficient exposure, the catheter tip was clamped and retained within the bladder for one hour. The catheter was then removed, and the procedure concluded. Antimicrobial susceptibility of the *A. baumannii* isolate was determined using the VITEK 2 system with GN ID and AST N420 cards (bioMérieux, Marcy-l'Étoile, France). The isolate was resistant to piperacillin–tazobactam, imipenem, meropenem, amikacin, ciprofloxacin, levofloxacin, and trimethoprim–sulfamethoxazole, but susceptible to tigecycline. Tigecycline susceptibility was confirmed via E-test (Bioanalyse, Ankara, Turkiye), with a susceptibility threshold of ≤2 μg/mL as defined by the FDA for Enterobacterales (FDA) based on previous studies [47,48]

The antimicrobial susceptibility testing method used (automated broth microdilution method confirmed by E-test). Antimicrobial susceptibility was determined using the automated broth microdilution method and confirmed via E-test (bioMérieux). And the bacterial preparation protocol was used, including culture in Mueller–Hinton broth and turbidity adjustment using a densitometer to reach 5.0 McFarland standard (~1.5 × 10^9^ CFU/mL). For experimental inoculation, the isolate was subcultured in Mueller–Hinton broth, incubated at 37 °C for 18 h, and adjusted to a turbidity of 5.0 McFarland standard using a densitometer (DEN-1B, Biosan, Riga, Latvia), corresponding to an approximate bacterial density of 1.5 × 10^9^ CFU/mL.

Three days post-inoculation, the cystitis model was established. During this period, the rats were monitored without treatment, and their weight and general condition were assessed daily. No mortality occurred during this stage. Urine samples were collected under sterile conditions and cultured on 5% sheep blood agar and EMB agar using a quantitative method. Plates were incubated at 37 °C for 24 h. The growth of non-fermenting, oxidase-negative, Gram-negative coccobacilli on the culture media confirmed the presence of *A. baumannii*, with colony counts exceeding 10⁵ CFU/mL. 

Seven rats failed to develop a cystitis model, and five additional rats died during the study. These 12 rats were excluded, leaving 24 rats for the experimental phase.

### 4.3. Study Groups

The remaining 24 rats were randomly divided into four groups of 6:

Control group (untreated): Rats were only monitored without any treatment.

Intravesical saline group: Rats received 1 mL of 0.9% isotonic saline intravesically for five consecutive days.

Low-dose tigecycline group: Rats received intravesical instillation of 6.25 mg/kg/mL tigecycline for five days.

High-dose tigecycline group: Rats received intravesical instillation of 25 mg/kg/mL tigecycline for five days.

Anesthesia was used for bacterial inoculation, urine sampling, and sacrification. Xylazine (8 mg/kg, i.p.) and ketamine (70 mg/kg, i.p.) were administered as anesthetic agents.

### 4.4. Sacrifice and Sample Collection

Rats were euthanized by cervical dislocation, and bladder tissues were collected for histopathological and cytogenetic analysis. Tissue for histopathology was preserved in 10% formaldehyde, while tissue for cytogenetic analysis was stored at −80 °C in isotonic saline.

### 4.5. Microbiological and Histopathological Examination

Urine samples were cultured on 5% sheep blood and EMB agar. Colony growth was assessed after 18–24 h of incubation at 37 °C. Histopathological analysis of bladder tissues included assessment for inflammation, fibrosis, vascular proliferation, edema, ulceration, erosion, and epithelial regeneration.

### 4.6. Cytogenetic Analysis

DNA damage was evaluated using the Comet assay, a sensitive technique for detecting low levels of DNA damage. A total of 200 cells per sample were analyzed under fluorescence microscopy, and parameters such as tail length, tail moment, and tail density were recorded.

### 4.7. Statistical Analysis

The microbiological and histopathological data obtained in this study were analyzed using the Jamovi v.1.6 statistical software (Jamovi Project, Version 1.6, Sydney, Australia). Differences between groups were assessed using the Pearson Chi-Square test, with statistical significance set at *p* < 0.05.

To evaluate the normality of numerical variables, the Kolmogorov–Smirnov test, histogram visualization, and kurtosis-skewness coefficient values were employed. These analyses were conducted using the same statistical software. The Comet test results for genotoxicity were also analyzed by comparing control groups with treatment groups.

If the normality condition was met, the Independent Samples t-test was used. When normality was not satisfied, the Mann–Whitney U test was applied. A significance threshold of *p* < 0.05 was maintained throughout the analysis.

## 5. Conclusions

This study represents one of the first in vivo investigations to comprehensively evaluate the therapeutic efficacy of intravesical tigecycline administration in an experimental cystitis model induced by extensively drug-resistant (XDR) *Acinetobacter baumannii*. Microbiologically, intravesical tigecycline demonstrated significant bacterial clearance by day 5 of treatment, indicating its potential as an alternative to systemic therapy (*p =* 0.028). Histopathological analysis revealed a marked inflammatory response in the high-dose group, highlighting dose-related tissue effects of local administration (*p =* 0.029). Furthermore, the Comet assay results showed evidence of genotoxicity in both tigecycline groups, suggesting that such effects may occur independently of dose (*p <* 0.05).

Based on these findings, intravesical tigecycline may offer a promising approach for treating resistant urinary tract infections. However, due to the observed inflammation and potential genotoxic effects, further investigation is warranted before clinical application. Larger-scale, long-term studies exploring various dosing protocols are recommended to validate the efficacy and safety of this treatment strategy. As a result of further studies, it is predicted that it may be an option in the treatment of many patients, to reduce mortality, morbidity, and hospitalization times in patients, and indirectly reduce healthcare-related costs as a result.

## Figures and Tables

**Figure 1 antibiotics-14-00611-f001:**
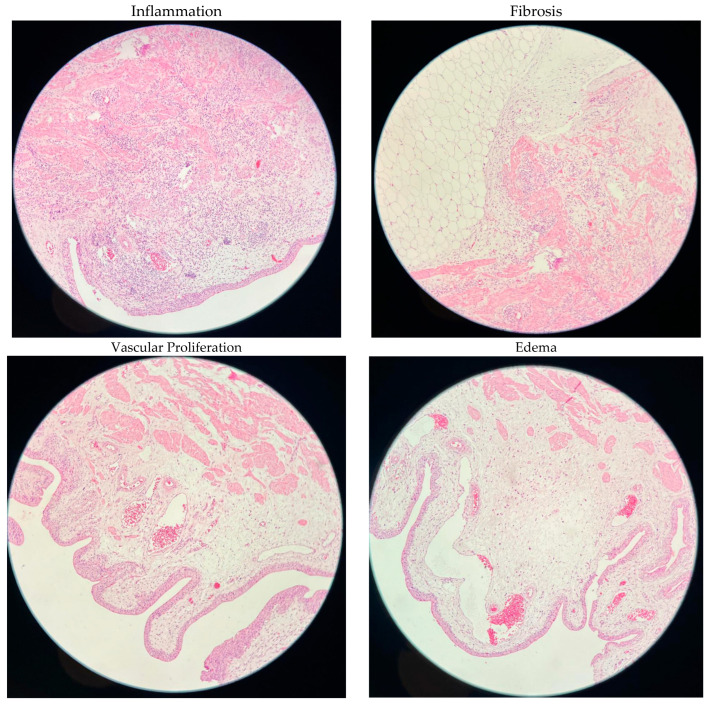
Histopathological figures.

**Figure 2 antibiotics-14-00611-f002:**
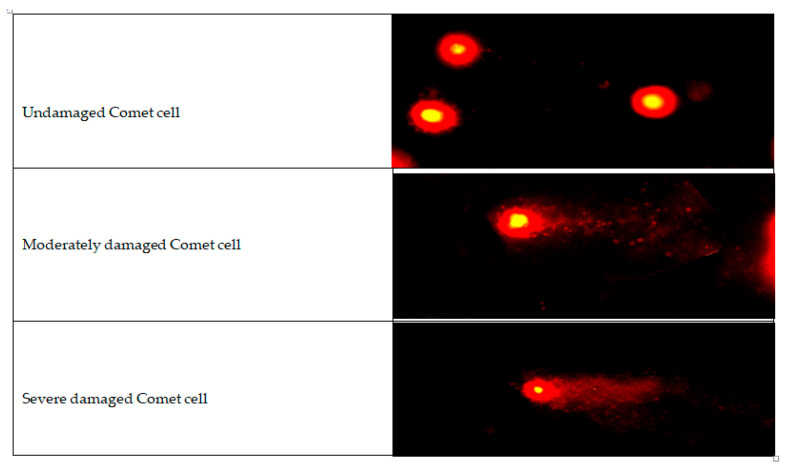
Comet cell on electrophoresis.

**Figure 3 antibiotics-14-00611-f003:**
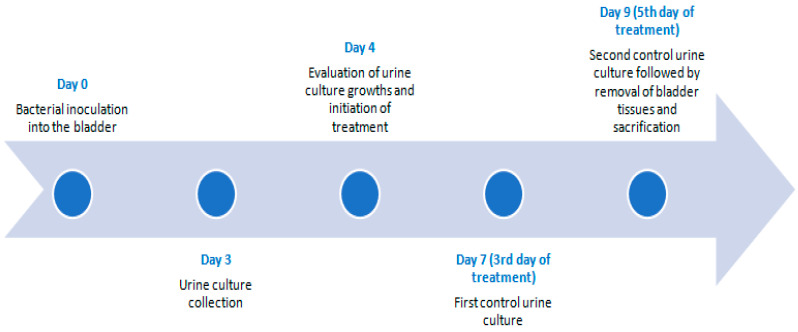
Chronological experiment chart.

**Table 1 antibiotics-14-00611-t001:** The urine culture results for each group on the 3rd and 5th day of treatment.

	3rd Day of Treatment (6th Day of Infection)		*p*	5th Day of Treatment (8th Day of Infection)		** *p* **
	Culture-Negative	Culture-Positive		Culture-Negative	Culture-Positive	
	*n* (%)	*n* (%)		*n* (%)	*n* (%)	
Untreated	0 (0)	6 (100)	0.082	0 (0)	6 (100)	0.028
Saline Solution	2 (33.3)	4 (66.7)		2 (33.3)	4 (66.7)	
Low-Dose Tigecycline	4 (66.7)	2 (33.3)		4 (66.7)	2 (33.3)	
High-Dose Tigecycline	4 (66.7)	2 (33.3)		5 (83.3)	1 (16.7)	

**Table 2 antibiotics-14-00611-t002:** Histopathological findings.

Histopathological Findings	Percentage of Area	Study Groups				
		Untreated	Saline Solution	Low-Dose Tigecycline	High-Dose Tigecycline	*p*
		*n* (%)	*n* (%)	*n* (%)	*n* (%)	
Inflammation	0%	4 (66.7)	2 (33.3)	2 (33.3)	2 (33.3)	0.029
	<25%	1 (16.7)	4 (66.7)	4 (66.7)	1 (16.7)	
	25–50%	1 (16.7)	0 (0)	0 (0)	0 (0)	
	>50%	0 (0)	0 (0)	0 (0)	3 (50)	
Fibrosis	0%	4 (66.7)	3 (50)	3 (50)	2 (33.3)	0.308
	<25%	2 (33.3)	3 (50)	3 (50)	1 (16.7)	
	25–50%	0 (0)	0 (0)	0 (0)	2 (33.3)	
	>50%	0 (0)	0 (0)	0 (0)	1 (16.7)	
Vascular Proliferation	0%	4 (66.7)	2 (33.3)	2 (33.3)	2 (33.3)	0.220
	<25%	1 (16.7)	2 (33.3)	4 (66.7)	1 (16.7)	
	25–50%	1 (16.7)	2 (33.3)	0 (0)	1 (16.7)	
	>50%	0 (0)	0 (0)	0 (0)	2 (33.3)	
Edema	0%	4 (66.7)	1 (16.7)	0 (0)	2 (33.3)	0.114
	<25%	2 (33.3)	3 (50)	5 (83.3)	1 (16.7)	
	25–50%	0 (0)	2 (33.3)	1 (16.7)	2 (33.3)	
	>50%	0 (0)	0 (0)	0 (0)	1 (16.7)	
Ulceration	0%	5 (83.3)	5 (83.3)	5 (83.3)	3 (50)	0.759
	<25%	1 (16.7)	1 (16.7)	1 (16.7)	2 (33.3)	
	25–50%	0 (0)	0 (0)	0 (0)	1 (16.7)	
	>50%	0 (0)	0 (0)	0 (0)	0 (0%)	
Erosion	0%	6 (100)	4 (66.7)	5 (83.3)	6 (100)	0.573
	<25%	0 (0)	1 (16.7)	1 (16.7)	0 (0)	
	25–50%	0 (0)	1 (16.7)	0 (0)	0 (0)	
	>50%	0 (0)	0 (0)	0 (0)	0 (0)	
Epithelial Regeneration	0%	6 (100)	4 (66.7)	5 (83.3)	4 (66.7)	0.695
	<25%	0 (0)	2 (33.3)	1 (16.7)	1 (16.7)	
	25–50%	0 (0)	0 (0)	0 (0)	1 (16.7)	
	>50%	0 (0)	0 (0)	0 (0)	0 (0)	

**Table 3 antibiotics-14-00611-t003:** Genotoxicity evaluations.

	Tail Length (µm)	Tail Moment	Tail Intensity (%)	*p*
Untreated	13.13 ± 1.36	9.78 ± 1.25	229.31 ± 0.53	<0.05
Saline (SF)	14.08 ± 1.55	10.69 ± 1.48	230.84 ± 0.61	
Low-Dose TGC	18.47 ± 1.88	15.61 ± 1.89	233.34 ± 0.71	
High-Dose TGC	21.24 ± 2.11	17.99 ± 2.12	233.89 ± 0.71	

## Data Availability

Data available upon request.

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
