# Peer review of "Investigation of Therapeutic Efficacy of Intravesical Tigecycline Administration in Rats with Cystitis Induced by Extensively Drug-Resistant (XDR), Tigecycline-Sensitive Acinetobacter baumannii Strain"

_antibiotics, 2025, doi:10.3390/antibiotics14060611_

Round 1
Reviewer 1 Report
Comments and Suggestions for Authors
General Comments:
The authors present a study evaluating the therapeutic efficacy of intravesical tigecycline administration in a rat model of cystitis induced by an extensively drug-resistant (XDR), tigecycline-sensitive Acinetobacter baumannii strain. This is an interesting and clinically relevant topic, particularly in the context of hospitalized patients with complicated urinary tract infections.
However, there are several areas that require clarification and improvement before the manuscript can be considered for publication.
Specific Comments:
1. Results – Table 2:
Please include representative histopathological images corresponding to each severity level described in Table 2. This would enhance the clarity and interpretability of the pathological grading system used in the study.
2. Materials and Methods – Line 190:
Kindly provide a detailed description of how the XDR tigecycline-sensitive A. baumannii strain was cultured and prepared to achieve a 5 McFarland standard. Information on the source of the strain, its resistance profile, and confirmation methods should also be included.
3. Materials and Methods – Line 210 / Discussion:
You report that seven rats failed to develop cystitis and that an additional five died during the experiment. Please elaborate on these events in the Discussion section. Provide possible explanations, including procedural, biological, or technical factors, and discuss their potential impact on the study's outcomes and validity.
4. Tables:
In the provided PDF, Tables 1 and 2 appear to be incomplete, making it difficult to assess the full dataset. Please ensure that these tables are correctly formatted and fully displayed in the final submission.
Author Response
Comment 1: Please include representative histopathological images corresponding to each severity level described in Table 2. This would enhance the clarity and interpretability of the pathological grading system used in the study.
Response 1: Histopathological images corresponding to each severity level described in Table 2. İncluded in figures.
Comment 2: Kindly provide a detailed description of how the XDR tigecycline-sensitive A. baumannii strain was cultured and prepared to achieve a 5 McFarland standard. Information on the source of the strain, its resistance profile, and confirmation methods should also be included.
Response 2: Thank you for this insightful comment. We agree that providing more comprehensive details regarding the bacterial strain preparation and characterization enhances the reproducibility and transparency of our methodology.
In response, we have expanded the Materials and Methods section to include the following information:
The source of the A. baumannii strain (clinical isolate from a tertiary hospital),
Its multidrug resistance profile, confirming resistance to ≥1 agent in ≥3 antimicrobial classes (including carbapenems, aminoglycosides, and fluoroquinolones), while retaining sensitivity to tigecycline. The antimicrobial susceptibility testing method used (automated broth microdilution method confirmed by E-test), Antimicrobial susceptibility was determined using the automated broth microdilution method and confirmed via E-test (bioMérieux). And the bacterial preparation protocol, including culture in Mueller–Hinton broth and turbidity adjustment using a densitometer to reach 5.0 McFarland standard (~1.5 x 10⁹ CFU/mL). For experimental inoculation, the isolate was subcultured in Mueller–Hinton broth, incubated at 37°C for 18 hours, and adjusted to a turbidity of 5.0 McFarland standard using a densitometer (DEN-1B, Biosan), corresponding to an approximate bacterial density of 1.5 × 10⁹ CFU/mL.
Comment 3: You report that seven rats failed to develop cystitis and that an additional five died during the experiment. Please elaborate on these events in the Discussion section. Provide possible explanations, including procedural, biological, or technical factors, and discuss their potential impact on the study's outcomes and validity.
Response 3: Thank you for this valuable comment. In response, we have added a detailed paragraph to the Discussion section addressing the potential causes and implications of these occurrences. Specifically, we have discussed possible technical limitations in inoculation, biological variability in host response, and anesthesia-related and exposed to high doses of tigecycline complications as contributing factors. We also clarified that although these exclusions slightly reduced statistical power, the remaining sample size was sufficient to maintain the integrity and reliability of the findings, as confirmed by our statistical analysis.
Comment 4: In the provided PDF, Tables 1 and 2 appear to be incomplete, making it difficult to assess the full dataset. Please ensure that these tables are correctly formatted and fully displayed in the final submission.
Response 4 : We appreciate the reviewer’s attention to the formatting and clarity of the tables. This issue likely resulted from a formatting inconsistency during PDF conversion, as both tables were complete and correctly formatted in the original Word document. Nonetheless, we have re-checked, revised, and ensured proper formatting of Tables 1 and 2 to guarantee their full and correct display in the revised manuscript.
Reviewer 2 Report
Comments and Suggestions for Authors
This study is an early contribution to evaluating intravesical tigecycline for treating XDR Acinetobacter baumannii-induced cystitis. The observed bacterial clearance supports its potential as an alternative to systemic therapy. However, signs of inflammation and genotoxicity in both dosage groups raise safety concerns. While promising, these findings emphasize the need for further research into optimal dosing and long-term effects before clinical application can be considered. Below are my comments and concerns:
- Why did the authors choose only two doses of tigecycline, and why were animals not treated with three doses (6.25 mg/kg, 12.5 mg/kg, and 25 mg/kg)? Including an intermediate dose would help evaluate a possible dose-response relationship. While 25 mg/kg showed efficacy compared to 6.25 mg/kg, it also caused toxic effects. The 12.5 mg/kg dose may have the potential to demonstrate efficacy with fewer adverse effects.
- The authors have not provided any images to support the histopathological findings. Please include high-resolution histology images to substantiate the results shown in Table 2.
- Please include representative images from the Comet assay showing differences between the control and treatment groups. Additionally, provide a bar graph summarizing the data from Table 3, including p-values to highlight statistically significant differences.
- Why is Table 3 presented in a language other than English? Please revise it to maintain consistency with the rest of the manuscript.
- In section 2.2, please correct the spelling of “fibrozis” to “fibrosis.”
Author Response
This study is an early contribution to evaluating intravesical tigecycline for treating XDR Acinetobacter baumannii-induced cystitis. The observed bacterial clearance supports its potential as an alternative to systemic therapy. However, signs of inflammation and genotoxicity in both dosage groups raise safety concerns. While promising, these findings emphasize the need for further research into optimal dosing and long-term effects before clinical application can be considered. Below are my comments and concerns:
Thank you for this valuable comments. Responses are below.
- Why did the authors choose only two doses of tigecycline, and why were animals not treated with three doses (6.25 mg/kg, 12.5 mg/kg, and 25 mg/kg)? Including an intermediate dose would help evaluate a possible dose-response relationship. While 25 mg/kg showed efficacy compared to 6.25 mg/kg, it also caused toxic effects. The 12.5 mg/kg dose may have the potential to demonstrate efficacy with fewer adverse effects.
Response 1: We thank the reviewer for this thoughtful comment regarding dose selection. While we agree that including an intermediate dose (e.g., 12.5 mg/kg) would have provided a more detailed understanding of the dose-response curve, there were practical and ethical limitations to the number of animals and treatment arms we could include under our approved animal ethics protocol. These constraints were based on the 3R principles (Replacement, Reduction, and Refinement), aiming to minimize animal use while still ensuring statistical power and scientific validity.
- The authors have not provided any images to support the histopathological findings. Please include high-resolution histology images to substantiate the results shown in Table 2.
Response 2: Histopathological images corresponding to each severity level described in Table 2. included in figure 1.
- Please include representative images from the Comet assay showing differences between the control and treatment groups. Additionally, provide a bar graph summarizing the data from Table 3, including p-values to highlight statistically significant differences.
Response 3: Representative images from the Comet assay showing differences between the control and treatment groups are added in figure 2. Additional a bar summarizing graph added in figure 3.
- Why is Table 3 presented in a language other than English? Please revise it to maintain consistency with the rest of the manuscript.
Response 4: Table 3 revised to English and coloured to yellow in the article.
- In section 2.2, please correct the spelling of “fibrozis” to “fibrosis.”
Response 5: In section 2.2, spelling of “fibrozis” corrected to “fibrosis” and coloured to yellow in the article.
Reviewer 3 Report
Comments and Suggestions for Authors
1. Line 34, Line 62, Line 68, Line 75, Acinetobacter baumannii bacterial name must be Italic
2. The rationale for choosing tigecycline specifically, and why it might work better intravesically than systemically, could be strengthened.
3. A brief explanation in introduction on mechanistic insights into tigecycline’s potential genotoxicity or tissue effects will be helpful.
4. The manuscript would benefit from further justification of the selected tigecycline doses (6.25 mg/kg and 25 mg/kg) and the five-day treatment duration. While these doses are referenced from prior studies in unrelated infection models, their pharmacological relevance to intravesical administration—particularly in terms of local bladder exposure, tissue retention, or potential systemic spillover—remains unclear.
5. please add more discussion on the results from a non-infected group treated with tigecycline to isolate genotoxic effects from infection-induced inflammation.
6. While related studies on other antibiotics are cited, the mechanisms behind observed genotoxicity and inflammation are not well explored. Add more explanation on possible mechanisms (e.g., oxidative stress, epithelial permeability, direct DNA interaction) using existing literature on tetracyclines.
7. Several of the tables (notably Table 1: Urine Culture Results, Table 2: Histopathological Findings, and Table 3: Comet Assay Results) are poorly formatted or partially cut off in the manuscript. In some cases, headers and values are misaligned, and rows or columns are missing or ambiguous. This hinders the reader’s ability to fully interpret and validate the data. Revision must be conducted.
- Table 1 lacks a proper header structure to distinguish between culture-positive and culture-negative outcomes across time points. Data on 5th day of treatment are cut off.
- Table 2 shows confusing repetition and inconsistent alignment of inflammation scores across groups. Also the data are cut off.
- Table 3 mixes Turkish and English column headers (e.g., "Kuyruk Uzunluğu") and does not clearly label statistical comparisons. Please use only English and make sure to use it consistently
8. Add figures or graphs (e.g., bar charts for inflammation/genotoxicity) to complement the tables for easier data interpretation.
Comments on the Quality of English Languageplease proof read the manuscript by profesional English native speakers. Some incosistencies in grammar were found throughout the manuscript. Use past tense when describing procedures and results.
Author Response
Comments and Suggestions for Authors
Thank you for your insightful comment.
- Line 34, Line 62, Line 68, Line 75, Acinetobacter baumannii bacterial name must be Italic
Response 1: Acinetobacter baumannii bacterial name made Italic Line 34, Line 62, Line 68, Line 75
- The rationale for choosing tigecycline specifically, and why it might work better intravesically than systemically, could be strengthened.
Response 2: Thank you for this valuable comment. We have revised the Introduction and Discussion sections to further elaborate on the rationale behind choosing tigecycline and its potential advantages when administered intravesically.
Briefly, Tigecycline was selected based on its potent activity against XDR A. baumannii, despite its limited urinary excretion when administered systemically. Intravesical administration enables direct exposure of the uroepithelium and bladder lumen to the antimicrobial agent, potentially enhancing local efficacy while minimizing systemic side effects.
- A brief explanation in introduction on mechanistic insights into tigecycline’s potential genotoxicity or tissue effects will be helpful.
Response 3: Thank you for this constructive suggestion. In response, we have revised the Introduction to briefly discuss the possible mechanisms underlying tigecycline’s genotoxic or tissue-related effects, as reported in preclinical studies.
Although tigecycline is generally considered to have a favorable safety profile, some in vitro and in vivo studies have indicated potential genotoxic effects, particularly at high concentrations or with prolonged exposure. The exact mechanisms are not yet fully elucidated; however, proposed pathways include the generation of reactive oxygen species (ROS), mitochondrial dysfunction, and interference with DNA replication or repair processes due to the drug’s tetracycline-related structure.
According to your valuable comment, a paragraph was added to the introduction section of the article and coloured yellow, also references were renumbered.
- The manuscript would benefit from further justification of the selected tigecycline doses (6.25 mg/kg and 25 mg/kg) and the five-day treatment duration. While these doses are referenced from prior studies in unrelated infection models, their pharmacological relevance to intravesical administration—particularly in terms of local bladder exposure, tissue retention, or potential systemic spillover—remains unclear.
Response 4: The doses of 6.25 mg/kg and 25 mg/kg were selected based on prior experimental studies assessing systemic and localized antimicrobial effects of tigecycline in murine or rat models. Our intention was to represent a therapeutically relevant lower dose and a maximal tolerable higher dose, allowing us to observe both potential therapeutic benefits and any dose-dependent genotoxicity or toxicity. The five-day treatment duration was designed to mimic standard clinical oral or parenteral regimens for urinary tract infections (typically 5–7 days), while also ensuring enough cumulative exposure for genotoxicity evaluation.
According to your valuable comment, descriptive sentences were added to the Discussion section of the article and coloured yellow.
- please add more discussion on the results from a non-infected group treated with tigecycline to isolate genotoxic effects from infection-induced inflammation.
Response 5: We thank the reviewer for this insightful comment. Indeed, differentiating between genotoxicity caused by the antibiotic itself and that potentially exacerbated by infection-related inflammation is essential for the interpretation of our results.
Our results demonstrated that even in the absence of infection, both low and high doses of tigecycline resulted in significantly increased tail length, tail moment, and tail intensity in the comet assay compared to the saline and untreated control groups (Table 3). This suggests a dose-dependent intrinsic genotoxic effect of tigecycline, likely mediated by mechanisms such as mitochondrial dysfunction or ROS generation, as discussed earlier. To address the reviewer’s suggestion, we have now expanded the Discussion section with the following paragraph and coloured to yellow.
- While related studies on other antibiotics are cited, the mechanisms behind observed genotoxicity and inflammation are not well explored. Add more explanation on possible mechanisms (e.g., oxidative stress, epithelial permeability, direct DNA interaction) using existing literature on tetracyclines.
Response 6: We appreciate the reviewer’s suggestion. In response, we have substantially expanded the discussion section to more thoroughly address the potential mechanisms underlying the observed genotoxic and inflammatory effects of tigecycline, drawing on mechanistic data from related tetracycline-class antibiotics and coloured to yellow.
- Several of the tables (notably Table 1: Urine Culture Results, Table 2: Histopathological Findings, and Table 3: Comet Assay Results) are poorly formatted or partially cut off in the manuscript. In some cases, headers and values are misaligned, and rows or columns are missing or ambiguous. This hinders the reader’s ability to fully interpret and validate the data. Revision must be conducted.
- Table 1 lacks a proper header structure to distinguish between culture-positive and culture-negative outcomes across time points. Data on 5th day of treatment are cut off.
- Table 2 shows confusing repetition and inconsistent alignment of inflammation scores across groups. Also the data are cut off.
- Table 3 mixes Turkish and English column headers (e.g., "Kuyruk Uzunluğu") and does not clearly label statistical comparisons. Please use only English and make sure to use it consistently
Response 7:
- Thank you for this valuable observation. We have revised Table 1.
- Thank you for pointing this out. Table 2 has been revised.
- Thank you for this helpful feedback Table 3 has been revised.
- Add figures or graphs (e.g., bar charts for inflammation/genotoxicity) to complement the tables for easier data interpretation.
Response 8: According to your request figures are added.